

# Expression and molecular characterization of an intriguing hyaluronan synthase (HAS) from the symbiont "*Candidatus Mycoplasma liparidae*" in snailfish

Lulu Guo,  Shaolu Wang,  Chunang Lian and  Lisheng He

Department of Deep-sea Science Research, Institute of Deep-Sea Science and Engineering, Chinese Academy of Sciences, Sanya, Hainan, China

Corresponding author
Lisheng He, he-lisheng@idsse.ac.cn

## ABSTRACT

**Background**. Hyaluronan synthases (HASs) are ubiquitous in living organisms, and the hyaluronic acid (HA) synthesized by them are important to their body and well used in medicine, cosmetics and other fields. HAS from deep-sea creatures has not yet been explored before. The study aims to analyse the characteristics and enzyme kinetics of a novel hyaluronan synthase derived from the symbiont "*Candidatus* Mycoplasma liparidae" found in deep-sea snailfish (snHAS).

**Methodology**. snHAS was over-expressed using $His_6$ as tag in the study. The sequence alignment was conducted by Cluster $W$ and then the phylogenetic analyse of HASs was performed by Mega 6.0 to investigate the position of snHAS during evolution. $K_m$ and $V_{max}$ were detected to study the enzyme kinetics of snHAS wildtype and its mutant. The molecular weight of HA was evaluated by high performance gel permeation chromatography (HPGPC). The cardiolipin was added to investigate whether it had a promoting effect on the snHAS.

**Results**. The length of snHAS was 933 bp with an open reading frame (ORF) of 310 amino acids. Unlike other repoted HASs, snHAS had no transmembrane region and was not classified into the currently known Class I or Class II. snHAS could synthesize hyaluronan with lower molecular weights using the substrates of uridine-diphosphate—$N$-acetylglucosamine (UDP-GlcNAc) and uridine-diphosphate—glucuronic acid (UDP-GlcA) *in vitro*. The $K_m$ values of snHAS were $258 \pm 45$ μM and $39 \pm 5$ μM for UDP-GlcNAc and UDP-GlcA, respectively, much lower than those from mice ($K_m$ for UDP-GlcA: $55 \pm 5$ μM; $K_m$ for UDP-GlcNAc: $870 \pm 60$ μM). The $k_{cat}/K_m$ values of snHAS were $163.5$ s$^{-1}$ mM$^{-1}$ and $8.08$ s$^{-1}$ mM$^{-1}$ for UDP-GlcA and UDP-GlcNAc, respectively. Furthermore, the activity of snHAS was independent of cardiolipin.

**Conclusions**. snHAS was a novel HAS based on the characteristics of the animo acid sequence, which could produce low molecular weight of HA with high efficiency. This provides a molecular basis for the biosynthesis of low molecular weight of HA.

## INTRODUCTION

Hyaluronic acid, abbreviated as HA, is a linear glycosaminoglycan with a high molecular weight, typically ranging from 10 to $10^4$ kDa. It is found in nearly all multicellular organisms and some microorganisms (*Weigel, 2002*). HA is an essential structural component in synovial fluid, cartilage, the vitreous humor of the eye, and skin. Additionally, it plays important roles in regulating cellular behavior and various biological processes, including blood vessel wall permeability, protein, water, and electrolyte diffusion and operation, as well as promoting wound healing (*Bourguignon et al., 2000*; *Sohara et al., 2001*). Currently, HA has a wide range of applications in the medical, pharmaceutical, cosmetic, and dietary fields (*Brown & Jones, 2005*; *Oe et al., 2016*). Different molecular weights (MW) of HA exhibit distinct characteristics (*Coleman et al., 2000*; *Fujii et al., 1996*). High molecular weight HA generally acts as an osmotic buffer, space filler, and viscoelastic structure. Intermediate-sized HA, typically ranging from 200 to 500 kDa, can provoke pro-inflammatory responses in various pathologies. Conversely, HA with low molecular weight promotes wound healing and helps prevent adhesion after closure (*Snetkov et al., 2020*).

Hyaluronan synthases (HASs) synthesize HA in various organisms (*Weigel, 2015*; *Weigel & DeAngelis, 2007*). Classified as glycosyltransferases, HASs have been identified in vertebrates and certain microbial species. The first discovery of HAS was made in *Streptococci* in 1993 (*DeAngelis, Papaconstantinou & Weigel, 1993*), followed by identifications in humans and other microbes. HAS enzymes are categorized into two classes based on their transmembrane domains, protein sequences, and domain structures (*DeAngelis et al., 1998*; *Weigel, 2015*). HASs of Class I are proteins with integral membrane with a single glycosyltransferase 2 (GT2) family module, commonly comprising four to six transmembrane regions and one to two membrane-associated domains. For instance, *Streptococcal* HAS possesses four transmembrane segments and two amphipathic membrane domains (*Heldermon, DeAngelis & Weigel, 2001*). Additionally, *Streptococcal* HAS is reported to feature a cytoplasmic domain housing a motif (D, D, Q$XX$RW), which is crucial for the activity of glycosyltransferases that catalyze the $\beta 1 \rightarrow 4$ linkage in HA synthesis (*Itano & Kimata, 1998*; *Yoshida et al., 2000*). In contrast, Class II HASs are peripheral membrane proteins that contain two GT2 family modules. To date, only one Class II HAS has been identified, derived from *Pasteurella multocida* (pmHAS) (*DeAngelis et al., 1998*). Each GT2 module in pmHAS is responsible for catalyzing a specific type of glycosidic linkage (*Kooy et al., 2013*; *Kooy et al., 2014*; *Weigel & DeAngelis, 2007*). Notably, pmHAS possesses a significantly longer amino acid sequence of 972 residues, compared to Class I HAS enzymes, which typically range from 417 to 588 amino acids.

The deep sea is different from the shallow sea, with environmental characteristics such as high pressure, no light, low temperature, and oligotrophic conditions. In order to adapt to the deep-sea environment, deep-sea organisms, and their associated organisms, have evolved functional genes that are different from those of shallow-sea organisms. Therefore, deep-sea organisms are a huge gene resource bank. For example, the the N-acetylneuraminate lyase from the gut symbiotic bacteria of the deep-sea giant isopod

has a much higher affinity for the substrate than other N-acetylneuraminate lyase reported from shallow-sea sources (*Wang et al., 2018*). The fluorescent protein from deep-sea *Cribrinopsis japonica* has a longer excitation peak than that from shallow water (*Tsutsui et al., 2016*). Some crustins derived from deep-sea blind shrimp often have unique structures and activities (*Le Bloa et al., 2020*; *Wang et al., 2021*; *Guo et al., 2021*). Given the reported uniqueness of genetic resources from deep-sea organisms, we hypothesize that the HAS derived from deep-sea environments may possess distinctive characteristics. Consequently, we have initiated an in-depth investigation into snHAS.

Obtaining HA in an effective and environmentally friendly manner remains a challenge, particularly for low molecular weight HA (*Qiu et al., 2021*). In this study, we first cloned the snHAS gene from the intestinal microbiome "*Candidatus* Mycoplasma liparidae" of the deep-sea snailfish, which does not belong to either Class I or Class II. The recombinant snHAS demonstrated the ability to synthesize lower molecular weight of HA *in vitro* when supplied with exogenous UDP-GlcNAc and UDP-GlcA.

## MATERIALS AND METHODS

### Plasmid construction

To obtain the exogenously expressed protein snHAS, we inserted the snHAS gene sequence into an exogenous expression vector. The genomic DNA extraction, sequencing, assembly, and annotation were performed according to our previously published paper (*Lian et al., 2020*). In brief, a genomic DNA library of the hadal snailfish gut was constructed. Based on the sequences of the annotated HAS, two pairs of primers were designed, which were listed in Table S1. The HAS gene was amplified using PCR with the following protocol: a 50 $\mu$l reaction mixture was prepared containing five ng of the previously prepared cDNA, four $\mu$l of 10 mM dNTPs, 10 $\mu$l 5$\times$ PCR buffer, 0.5 $\mu$l Primer STAR HS DNA Polymerase (Takara, Tokyo, Japan), two $\mu$l of 10 $\mu$M for each primer, and 32.5 $\mu$l of ddH$_2$O. The PCR program was as follows: 98 °C for 10 s for denaturation, 30 cycles of 98 °C for 10 s, 50 °C for 30 s, and 72 °C for 1 min, with a final extension at 72 °C for 10 min. A template of full-length snHAS was used when the E173D mutant was generated with primers listed in Table S1. A pET-28a(+) vector was ligated with the purified PCR product. Sequencing (BGI, Shenzhen, China) was used to identify the positive clones, and subsequently, the expression of the target gene was carried out in BL21 (DE3) competent cells.

### Expression and purification

The wildtype and mutant of His-snHAS were expressed in BL21 (DE3) cells. Purification of His-snHAS wildtype and mutant was performed following the manufacturer' protocol (Sangon, Shanghai, China). In summary, a media containing 50 $\mu$g/ml of LB was used to culture the *E. coli* BL21 (DE3), which has recombinant plasmid, at 37 °C for 8 h. LB broth was used to dilute the cultures with 1,000:1 and the cultures were subjected to further incubation. Once the OD$_{600}$ value of the cultures reached approximately 0.6, a final concentration of 0.5 mM isopropyl β-D-1-thiogalactopyranoside (IPTG; Sigma, Shanghai, China) was added to induce the expression of target gene. Cells were collected after induction for 16 h at 28 °C and were then lysed using an ultrasonicator in a binding buffer
(20 mM Tris–HCl, 10 mM imidazole, 500 mM NaCl) kept on ice. After ultrasonication, cell debris was removed by centrifugation at 8,000×g for 35 mins at 4 °C. The supernatant was retained and mixed with Ni-NTA 6FF Sefinose Resin on ice. Recombinant proteins were eluted after thorough washing with a washing buffer (500 mM NaCl, 10 mM imidazole, 20 mM Tris–HCl). 1×PBS was used to dialyze the purified proteins at 4 °C for 24 h with buffer replacement every 12 h, carried out three times in total. The Bradford method was employed to determine the protein concentration, using bovine serum albumin (BSA) as the standard. The purified proteins were combined with a 6×SDS gel-loading buffer. After boiling for 10 min at 100 °C, the mixture was resolved with 12% SDS-PAGE (sodium dodecyl sulfate-polyacrylamide gel electrophoresis). Coomassie brilliant blue R250 was used to stain the gels. Finally, the purified proteins were stored in aliquots at −80 °C or used immediately, unless specified otherwise.

## Hyaluronan synthase assay

The activity of snHAS and its mutant snHAS E173D was examined using the following procedures. The assay was performed in a 50 µl reaction containing 15 mM MgCl$_2$, 50 mM phosphate buffer saline (PBS), five mM DL-dithiothreitol (DTT), one mM UDP-GlcNAc, one mM UDP-GlcA, and 30 µM purified snHAS wildtype or its mutant. After incubating the mixture for 1 h at 37 °C, the reaction was stopped by boiling, and then the reaction was tested by an HA Detection Kit following the manufacture's protocol (Shanghai Enzyme-linked Biotechnology Co., Ltd, Shanghai, China). Reactions without HAS were used as the control. In brief, HA binding protein (HABP) and horse radish peroxidase (HRP) were applied as primary and secondary probes, respectively. After chromogenic reaction with TMB, the OD$_{450}$ was recorded using Varioskan LUX (Thermo Fisher Scientific, Waltham, MA, USA). Each assay was repeated at least three times in this study. One enzymatic unit was defined on the amount of enzyme required when one pmol of HA in 1 min was synthesized within the above reaction.

For the hyaluronidase treatment, the synthesis reaction was monitored at 5 mins intervals for the first 40 mins, followed by 10 mins intervals for another 20 mins. After this, the mixture was incubated with two µg/µl hyaluronidase at 37 °C for 1 h, and assessed at 5 mins intervals. The reaction without hyaluronidase was used as a control.

## Enzyme kinetic analysis

The $K_m$ values of snHAS and its mutant were determined by measuring enzyme activities in the presence of UDP-GlcNAc and UDP-GlcA at various concentrations. For the wildtype snHAS, 30 µM of studied enzyme protein was used, with UDP-GlcA at a fixed concentration of one mM, and UDP-GlcNAc at varied concentrations (0, 0.01, 0.05, 0.1, 0.2, 0.4, 0.6, 0.8, 1.0 mM) were used. Alternatively, UDP-GlcNAc was at a fixed concentration of one mM, and UDP-GlcA at varied concentrations (0, 0.01, 0.025, 0.05, 0.1, 0.25, 0.5 mM) were used. For the mutant snHAS E173D, 30 µM of the enzyme, UDP-GlcA at a fixed concentration of one mM, and UDP-GlcNAc at varied concentrations (0, 0.2, 0.4, 0.6, 0.8, 1.0 mM) were used. Alternatively, UDP-GlcNAc at a fixed concentration of one mM, and UDP-GlcA at varied concentrations (0, 0.2, 0.4, 0.6, 0.8, 1.0 mM) were used. A nonlinear least-square

fitting procedure was employed to calculate the kinetic parameters of $K_m$ and $V_{max}$, using the Michaelis–Menten equation and curve fitting software.

In the experiment assessing the effect of cardiolipin on snHAS, freshly prepared proteins were used in the reaction mixture (15 mM MgCl$_2$, 50 mM PBS, five mM DTT, one mM UDP-GlcNAc, one mM UDP-GlcA, 30 μM purified snHAS) along with various concentrations of cardiolipin (1.0, 2.0, 3.0 mM). The reaction was incubated for 1 h at 37 °C, and the activities were detected using the methods described above.

### The molecular weight (MW) of HA

After the synthesis reaction was completed, three volumes of ethanol was added to the reaction. The mixture was then stored for 1 h at 4 °C. Following this, it was subjected to centrifugation at 5,000 r/min for 15 mins at 4 °C. The supernatant was then washed with an equal volume ethanol salt solution (75% w/v ethanol and 25% w/v 0.2 M NaCl). After another round of centrifugation at 10,000 rpm/min for 15 mins at 4 °C, the supernatant was discarded, and the sediment was freeze-dried. The purified HA was used for molecular weight detection. Briefly, the purified HA (5 mg/ml) was prepared along with standards (5000-667800 Da; Sigma-Aldrich, St. Louis, MO, USA) and analyzed using High Performance Gel Permeation Chromatography (HPGPC) with a BRT105-104-102 connected gel column (8×300 mm) (BoRui Saccharide Biotech Co. Ltd, Shenzhen, China), A 0.05 M NaCl was utilized as mobile phase, and the flow rate was set to 0.6 ml/min. The detection was carried out using an RI-10A refractive index detector.

## RESULTS

### Characteristics of snHAS sequence

The gene encoding snHAS was isolated from the cDNA library of the symbiotic "*Candidatus* Mycoplasma liparidae" residing in the hadal snailfish. The snHAS gene spans 933 base pairs, encoding a protein with 310 amino acids (accession number WWS21520) with a molecular weight (MW) of 37 kDa. A comparative analysis of snHAS with other HASs from diverse organisms was conducted (Fig. 1). Results revealed that compared with the motif of Class I HASs typically composed of D, D, Q$XX$RW (*Kooy et al., 2013*; *DeAngelis & Zimmer, 2023*), only two residues, 88D and 208Q, were conserved in snHAS (Fig. 1A). In contrast to Class I HASs, Class II HAS typically contains two GT2 modules (Table 1), whereas snHAS harbors only one predicted GT2 module (Fig. 1). Additionally, snHAS exhibits two motifs, DGSTD (38-42aa) and D$X$DD (88-91aa), akin to those found in the second GT2 module of Class II HASs (Fig. 1B). Furthermore, while 2-6 "B($X_7$)B" HA binding motifs were predicted in HASs from Mycoplasma and other organisms, only one such motif (273-281aa) was identified in snHAS (Table 1). Notably, snHAS lacks any predicted transmembrane regions, differing from glycosyltransferases in other Mycoplasma species (0-1) and HASs in other organisms (2-7), including both microorganisms and multicellular organisms (Table 1). Moreover, snHAS exhibits the smallest MW of 37.2 kDa, contrasting with the MWs of other previous reported HASs, which range from 44.78 to 112 kDa (Table 1). The isoelectric point (pI) values of these HASs range from 7.28 to 9.08, with

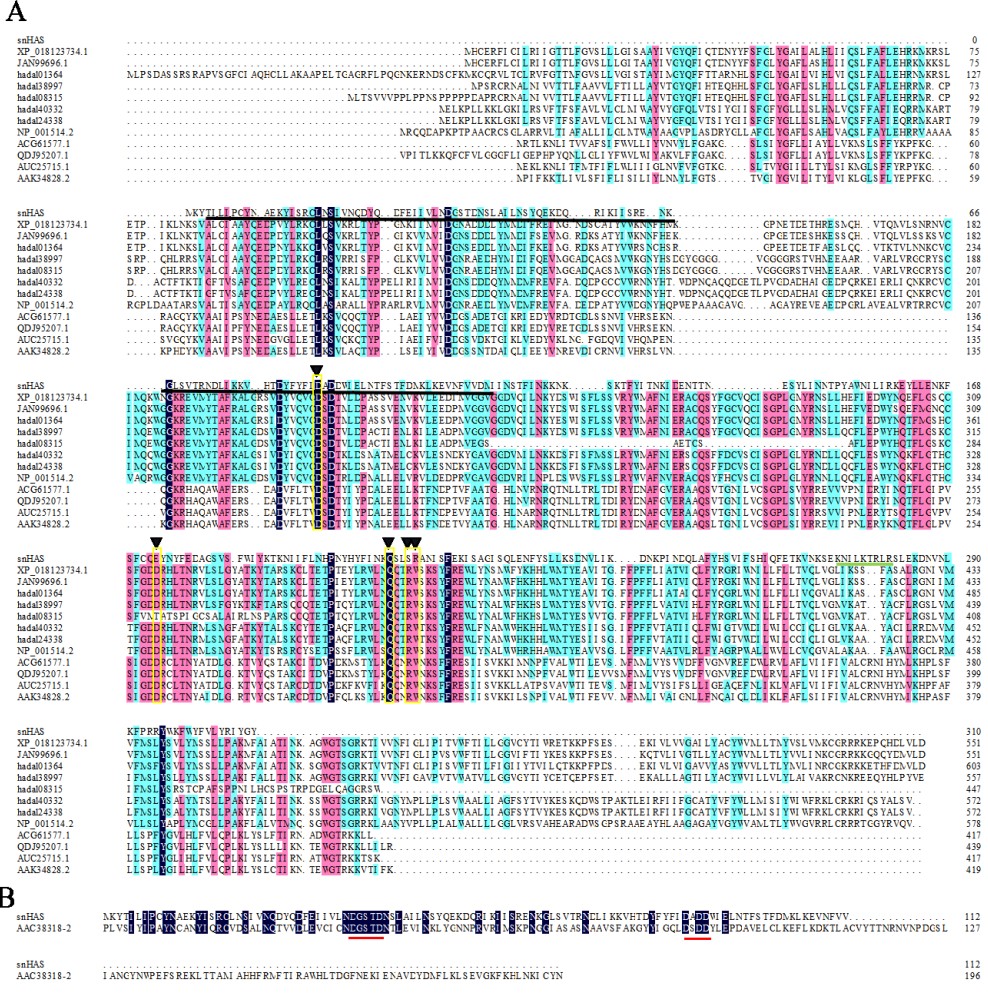

**Figure 1   Amino acid sequences alignments between snHAS and HAS from other organisms.** (A) The sequences used in this alignment were from *Xenopus laevis* (XP_018123734), *Heterocephalus glaber* (JAN99696), *Pseudoliparis swirei* (hadal01364, hadal38997, hadal08315, hadal24338), *Homo sapiens* (NP_001514), *Streptococcus equi* (ACG61577), *Streptococcus dysgalactiae* (QDJ95207), *Streptococcus uberis* (AUC25715), *Streptococcus pyogenes* (AAK34828). Triangles (▼) indicate the motif consisting of D, D, Q$XX$RW. The GT2 module of snHAS was underlined by solid black line. The HA binding motif of B($X_7$)B in snHAS was underlined by solid green line. (B) The GT2 module sequences from snHAS and pmHAS (AAC38318, *Pasturella multocida*) were used in the alignments. The two amino acid motifs consisting of DGSTD and D$X$ DD were underlined by solid red line. The sequences homology level was 100% which highlighted with black. The homology level of over 75% was highlighted with pink. The homology level of over 50% was highlighted with blue.

pmHAS displaying the lowest value of 7.28. Notably, no signal peptides were predicted in any of these HASs, including snHAS.

## Phylogenetic analysis of snHAS

To better understand the evolutionary relationship of snHAS with the glycosyltransferases from other organisms, a phylogenetic tree was constructed using sequences selected from

**Table 1 The number of predicted transmembrane regions, HA binding motifs, molecular weight, and pI.** A TMHMM Server v. 2.0 was used to evaluate the presence of transmembrane domains of the HASs.

| Organisms | GenBank No. | Class | Transmembrane regions | HA binding motifs | MW (molecular weight, kDa) | pI |
|---|---|---|---|---|---|---|
| *Candidatus* Mycoplasma | **WWS21520** | | **0** | **1** | **37.2** | **8.88** |
| | MDK2819671 | | 0 | 3 | 37.26 | 9.49 |
| Mycoplasma | MDR2568307 | – | 1 | 2 | 42.6 | 9.87 |
| | MDR2847095 | | 1 | 4 | 41.39 | 9.85 |
| *Xenopus laevis* | XP_018123734 | | 6 | 3 | 63.83 | 8.61 |
| *Heterocephalus glaber* | JAN99696 | | 6 | 3 | 63.47 | 8.74 |
| | PV421300 | | 7 | 3 | 68.91 | 9.08 |
| *Pseudoliparis swirei* | PV421301 | | 7 | 3 | 63.64 | 8.78 |
| | PV421302 | | 4 | 5 | 50.32 | 8.37 |
| | PV421303 | Class I | 7 | 3 | 66.74 | 8.45 |
| *Homo sapiens* | NP_001514 | | 7 | 3 | 64.84 | 9.34 |
| *Streptococcus equi* | ACG61577 | | 5 | 3 | 44.78 | 9.04 |
| *Streptococcus dysgalactiae* | QDJ95207 | | 5 | 3 | 50.35 | 9.06 |
| *Streptococcus uberis* | AUC25715 | | 5 | 4 | 47.33 | 9.32 |
| *Streptococcus pyogenes* | AAK34828 | | 5 | 4 | 47.92 | 8.82 |
| *Chlorella virus* | NP_048446.1 | | 7 | 2 | 65.17 | 8.24 |
| *Pasturella multocida* | AAC38318 | Class II | 2 | 6 | 112 | 7.28 |

**Notes.**
  Values in bold are from this study. A TMHMM Server v. 2.0 was used to evaluate the presence of transmembrane domains of the HASs.

the NCBI database. The results revealed that snHAS clustered closely withglycosyltransferases from the Mycoplasmatota bacterium (MDK289671, MDR2568307, MDR1235299), all of which belong to the Mycoplasmataceae family (Fig. 2). The sequence similarities to snHAS were 47.08%, 41.78% and 31.33% at the amino acid level. HASs of Class I were grouped together. The HASs from Mycoplasma formed a distinct cluster (Fig. 2).

## Catalytic activity

The catalytic activity of snHAS was examined using UDP-GlcNAc and UDP-GlcA as substrates. The results, presented in Fig. S1, demonstrated that snHAS could synthesize HA, with the amount of product positively correlated with the substrate concentrations prior to saturation. To verify the identity of the synthesized product, hyaluronidase was applied. The results showed that after treatment with hyaluronidase, the amount of HA decreased rapidly to almost undetectable levels (Fig. 3), confirming that the product synthesized was indeed hyaluronic acid. Subsequently, the enzymatic characteristics of snHAS were investigated. The results, displayed in Table 2, indicated that the $K_m$ value of snHAS for UDP-GlcNAc was $0.26 \pm 0.05$ mM, which was significantly lower than that of HAS from other organisms (Table 2). Meanwhile, the $K_m$ for UDP-GlcA was $0.04 \pm 0.01$ mM, comparable to enzymes from other sources (Table 2). For the $k_{cat}/K_m$ values, snHAS exhibited a higher value for UDP-GlcNAc compared to spHAS and a similar value to seHAS. Conversely, the $k_{cat}/K_m$ value of snHAS for UDP-GlcA was slightly lower than that of spHAS but comparable to seHAS (Table 2). The conserved motif D, D, Q$XX$RW in class

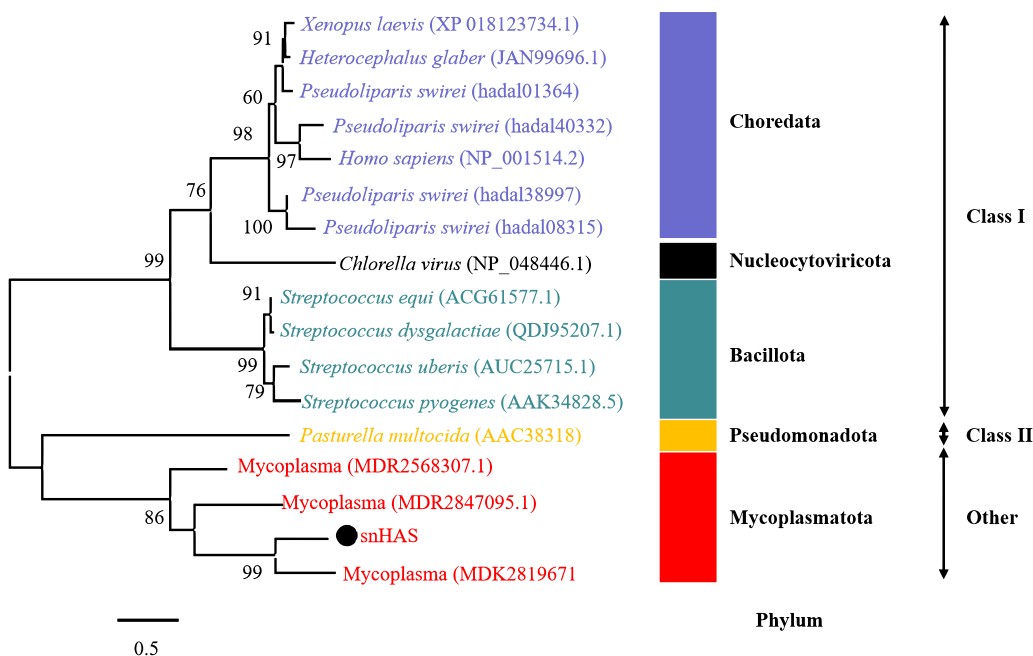

**Figure 2  Phylogenetic analysis of snHAS with HAS from other organisms.** The maximum likelihood tree was constructed using 17 glycosyltransferase sequences including snHAS with JTT matrix-based model. The bootstrap values were labeled at branches, indicating the percentage of times occurring in 1,000 replication by bootstrapping the original deduced protein sequences. snHAS in this study was marked by black dot. Sequences with the same color were from the same phylum.

I HAS was partially conserved in snHAS, with only the first D and Q being conserved. To investigate whether the residue "173D" is critical for the acticity of snHAS, we introduced a mutation that changed the amino acid E at the second D position to D, referred to as snHAS E173D mutant. The snHAS E173D mutant was cloned and overexpressed *in vitro*. Its activity was then examined and compared to that of the wild-type snHAS. The results showed that the snHAS E173D mutant retained HA synthesis activity. The $K_m$ values of snHAS E173D were $0.21 \pm 0.02$ mM for UDP-GlcA and $0.41 \pm 0.07$ mM for UDP-GlcNAc, representing increases of 1.26- and 5.25-fold over the wild-type, respectively. Additionally, the $k_{cat}/K_m$ value for the E173D mutant with UDP-GlcNAc was 5.9-fold higher than that of the wild-type, while the $k_{cat}/K_m$ value for UDP-GlcA was lower than that of the wild-type (Table 2).

## The effect of cardolipin

To explore the effect of cardiolipin on the activity of snHAS, cardiolipin was added to the reaction at final concentrations of one mM, two mM, and three mM. The results indicated that the activity of snHAS did not show significant changes when cardiolipin was present compared to the reaction that did not include cardiolipin (Fig. 4).

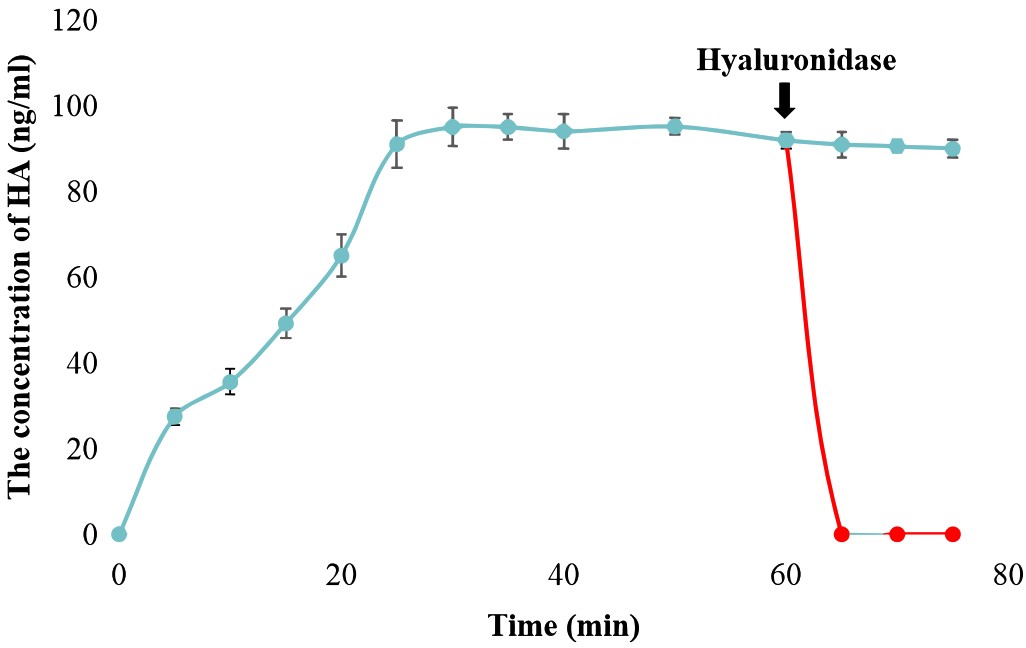

**Figure 3** **Examination of snHAS activity.** The HA was synthesized by snHAS and then treated with hyaluronidase. The progress was carried out following the methods outlined in Material and Methods. The points of the figure was shown as scatter plot. The reaction without hyaluronidase was depicted in blue; the reaction with hyaluronidase added at 60 mins was depicted in red.

## Measurement of HA molecular weight

To investigate the molecular weight of HA synthesized by snHAS, the molecular weight was examined by HPGPC. Results showed two distinct peaks. Upon comparison with the control, the molecular weight of HA synthesized by snHAS was approximately 3.5 kDa and 8.4 kDa (Fig. 5).

## DISCUSSION

Currently, HASs are classified into two classes: Class I and Class II, based on factors such as the number of GT2 modules and predicted topology (*DeAngelis, 1999*). Class I HASs are further subdivided into two subtypes: Class I-R, which has a reducing terminus, and Class I-N, which has a non-reducing terminus (*Weigel, Hascall & Tammi, 1997*; *Górniak et al., 2024*). Both Class I and Class II HASs are reported to contain 2–7 membrane domains, while snHAS have none. Other HASs from Mycoplasma typically have 0-1 membrane domain. In Class I HASs, two conserved sites have been identified, while Class II HASs have two conserved motifs found in snHAS. These correspond to the residues 88D and 208Q, and motifs DGSTD and D*X*DD (Fig. 1). Class I HASs commonly contain 4-18 Cys residues, whereas Class II HASs, represented by pmHAS, have 13 Cys residues. In contrast, snHAS has only three Cys residues, with two of them located in the GT2 module. Previous research has reported that two Cys residues are conserved in some Class I1 HASs, and one of these (Cys-225) in spHAS may be essential, as enzyme activity is partially inhibited when

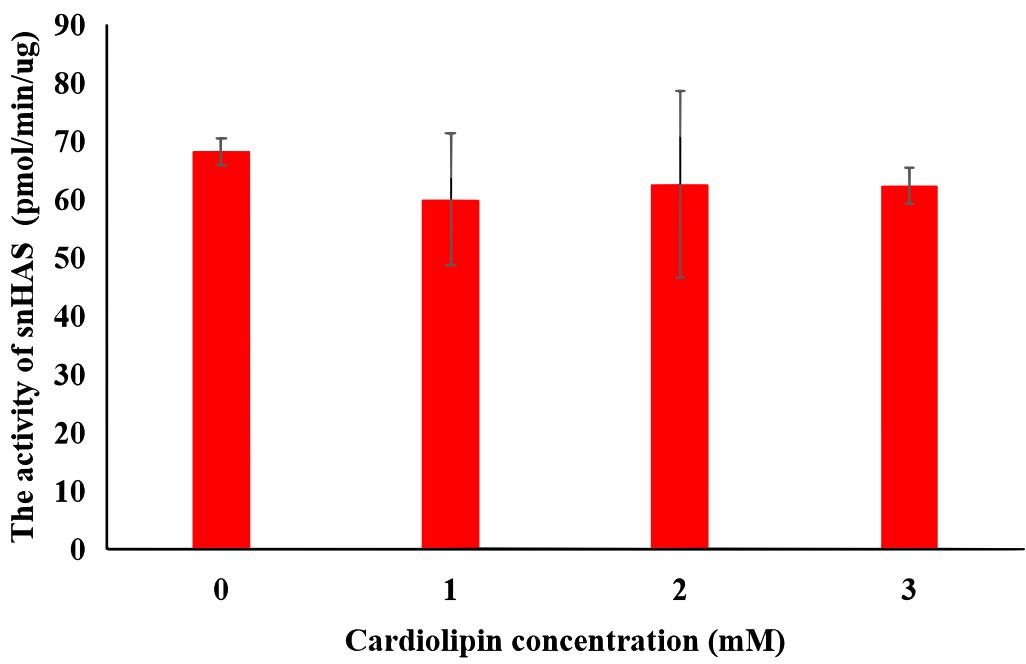

**Figure 4** **Effect of cardiolipin on the activity of purified snHAS.** Means ± SD of values are shown (standard deviation; N ≥ 3).

this Cys is modified (*Weigel, Hascall & Tammi, 1997*). The precise mechanism by which Cys residues affect enzyme activity remains unclear. Overall, the number of Cys residues in Class I and Class II HASs is greater than that in snHAS, suggesting that snHAS could represent a new type of HAS based on the analysis of its sequence characteristics.

Both Class I and Class II HASs possess transmembrane structures. They synthesize HA within the cytoplasm of cells, continuously elongating the HA sugar chain as it passes through the transmembrane domain before finally secreting it into the extracellular space (*Kaback, 2005*; *Tlapak-Simmons et al., 1999*). Most of the molecular weight of the HA they synthesize ranges from 10 to $10^4$ kDa. Studys indicate that the size of HA is influenced by the transmembrane structure, which not only facilitats HA transport but also aids in the chain formation (*Tlapak-Simmons et al., 1999*). The transmembrane channel is believed to stabilize the long chains of HA. Certain conserved amino acids located in the membrane domains (MDs) of Class I HASs have been shown to affect HA size. For instance, Lys48 in MD2 and Cys226 in MD3 in seHAS affect the size of HA synthesized by seHAS (*Kumari et al., 2006*; *Weigel & Baggenstoss, 2012*). In contrast, snHAS lacks a transmembrane structure as well as the specific conserved amino acids mentioned. This may explain why the molecular weight of HA synthesized by snHAS is considerably smaller than that produced by reported HASs. Another contributing factor could be that snHAS has only one HA binding site, unlike the HASs from other sources, leading to a less stable interaction between HA and snHAS and resulting in the premature release of HA before it can reach a sufficient length.

Guo et al. (2025), *PeerJ*, DOI 10.7717/peerj.19253

**Table 2  Michaelis constants for snHAS and other organisms.**

| | | | snHAS | snHASE173D | spHAS | seHAS | HAS1 |
|---|---|---|---|---|---|---|---|
| | $K_m$ | mM | $0.26 \pm 0.05$ | $0.41 \pm 0.07$ | $1.09 \pm 0.12$ | $0.95 \pm 0.17$ | $0.87 \pm 0.06$ |
| | $V_{max}$ | pmol/μ g/min | $62.9 \pm 2.6$ | $588.2 \pm 5$ | $50.0 \pm 21$ | $156.7 \pm 23$ | $24.2 \pm 2.1$ |
| UDP-GlcNAc | $k_{cat}$ | $s^{-1}$ | $2.1 \pm 0.09$ | $19.61 \pm 0.17$ | $1.66 \pm 0.7 - 2.77 \pm 1.16$ | $5.22 \pm 0.76 - 8.7 \pm 0.12$ | – |
| | $k_{cat}/K_m$ | $s^{-1} mM^{-1}$ | 8.08 | 47.8 | 1.52–2.54 | 5.49–9.15 | – |
| | $K_m$ | mM | $0.04 \pm 0.01$ | $0.21 \pm 0.02$ | $0.01 \pm 0.01$ | $0.04 \pm 0.02$ | $0.06 \pm 0.01$ |
| | $V_{max}$ | pmol/μ g/min | $196.1 \pm 29$ | $714.3 \pm 87$ | $58.3 \pm 18$ | $146.7 \pm 18$ | $21.3 \pm 1.9$ |
| UDP-GlcA | $k_{cat}$ | $s^{-1}$ | $6.54 \pm 0.97$ | $23.81 \pm 2.9$ | $1.94 \pm 0.6 - 3.23 \pm 1.0$ | $4.89 \pm 0.6 - 8.15 \pm 1.0$ | – |
| | $k_{cat}/K_m$ | $s^{-1} mM^{-1}$ | 163.5 | 113.38 | 194–323 | 122.25–203.75 | – |
| Organisms | | | "*Candidatus* Mycoplasma liparidae" | | *Streptococcus pygenes* | *Streptococcus equisimilis* | Mouse |
| Reference | | | This study | | *Tlapak-Simmons et al. (1999)* | | *Yoshida et al. (2000)* |

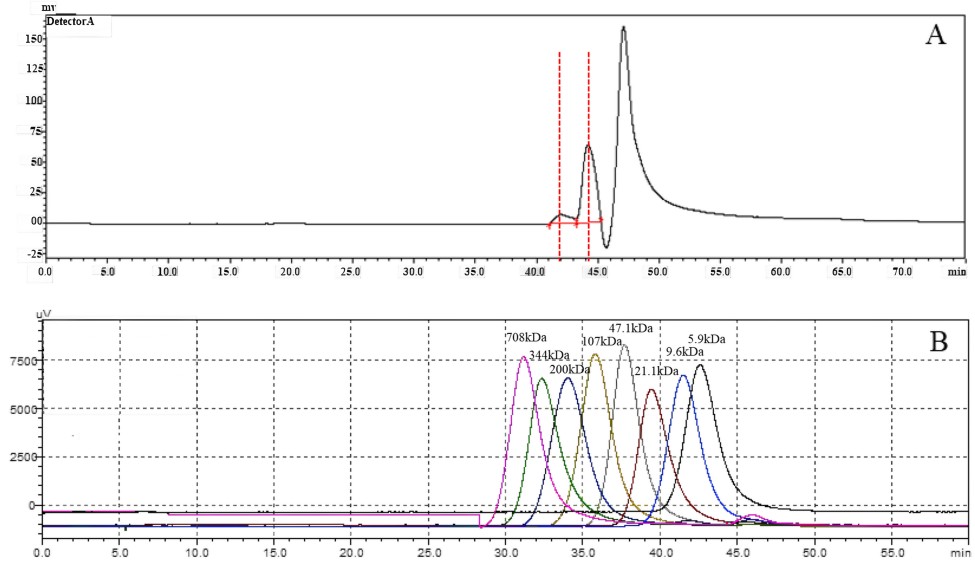

**Figure 5 The molecular weight analysis by using HPGLC.** (A) HA synthesized by snHAS. Two spectral peaks were detected. (B) Eight standard controls (5.9, 9.6, 21.1, 47.1, 107, 200, 344, 708 kDa respectively).

Previous studies have reported a significant effect of cardiolipin on the activity of Class I HASs (*Tlapak-Simmons et al., 1999*; *Tlapak-Simmons, Baron & Weigel, 2004*; *Weigel, Kyossev & Torres, 2006*). For both spHAS and seHAS, purifying them under cardiolipin (CL)—deficient conditions results in a substantial increase in enzyme activity upon the re-addition of CL (approximately 0.5-fold for spHAS and 1.5-fold seHAS). Moreover, the specific activity of these enzymes also rises with increasing CL concentration. However, no significant effect of CL was observed on snHAS in this study. One possible mechanism for cardiolipin's effect on HAS is that, during this process, HAS proteins interact with approximately 16 cardiolipin molecules to form a complex active enzyme. Cardiolipin engages with the cell membrane lipid bilayer in hydrophobic interactions, which aids in the transport of the HA sugar chain (*Tlapak-Simmons et al., 1999*; *Weigel, Kyossev & Torres, 2006*). In the absence of added cardiolipin, HAS demonstrates very low activity. In contrast, pmHAS is characterized a peripheral membrane protein and does not have an intrinsic affinity for lipid bilayers; therefore, its activity is independent of lipid (*Jing & DeAngelis, 2000*). The C-terminal region (amino acids 703-972aa) of pmHAS docks peripherally with an integral membrane transport protein in *Pasteurella multocida*, assisting in the transport of HA to extracellular space. In this study, snHAS does not exhibit a transmembrane structure. It is possible that the HA produced by snHAS does not require transport, which would render cardiolipin irrelevant for promoting snHAS activity. This suggests that snHAS may lack a transmembrane region.

Low molecular weight HA oligosaccharides ($\leq$10 kDa) are widely applied in healthcare, food, and medicine (*Boltje, Buskas & Boons, 2009*). Various methods have been explored to produce small molecules of HA, including physical degradation, chemical degradation, and biological degradation (*Qiu et al., 2021*). HA obtained through physical degradation

has a narrow molecular weight distribution but the process can be time-consuming (*Choi et al., 2010*; *Miyazaki, Yomota & Okada, 2001*). On the other hand, chemical degradation is quicker and more cost-effective; however, it can be harsh and environmentally unfriendly (*Hokputsa et al., 2003*; *Tommeraas & Melander, 2008*). Currently, HA oligosaccharides can only be effectively obtained through biological degradation (*Jin et al., 2016*). In this study, HA synthesized by snHAS has molecule weights of approximately 3.5 kDa and 8.4 kDa, making it suitable for direct application without further processing. This provides a versatile source for various future applications, such as promoting angiogenic, enhancing wound healing and improving cell adhesion. For instance, research has show that low molecular weight of HA facilitats endothelial cell proliferation, while high molecular weight of HA has the opposite effect (*Dovedytis & Samuel Bartlett, 2020*).

The comparative results of this study show that glycosyltransferases, including snHAS derived from Mycoplasma, differ in their transmembrane structures and molecular weights. HASs from other sources have more membrane domains and greater MWs compared to those from Mycoplasma. Typically, the snHAS analyzed in this study contain conserved sites from both Class I and Class II. This classification of snHAS may represent a novel type.

## CONCLUSIONS

The snHAS derived from the deep-sea symbiont Mycoplasma lacks transmembrane domains and is much smaller than previously reported HASs. *In vitro* study show that snHAS can effectively synthesize lower MW HA without the presence of cardiolipin, making it a promising source for future applications. Additionally, snHAS does not belong to either Class I or Class II HAS according to current classification standards. While current experiments indicate that snHAS can synthesize HA efficiently, further research is needed to examine and enhance its stability to meet the requirements for industrial production.

### Funding

This work was supported by the Major Scientific and Technological Projects of Hainan Province (NO. ZDKJ2021028) and Hainan Provincial Natural Science Foundation of China (NO. 322CXTD531). The funders had no role in study design, data collection and analysis, decision to publish, or preparation of the manuscript.

### Grant Disclosures

The following grant information was disclosed by the authors:
The Major Scientific and Technological Projects of Hainan Province: NO. ZDKJ2021028.
Hainan Provincial Natural Science Foundation of China: NO. 322CXTD531.

### Competing Interests

The authors declare there are no competing interests.

## Author Contributions

- Lulu Guo conceived and designed the experiments, performed the experiments, analyzed the data, prepared figures and/or tables, authored or reviewed drafts of the article, and approved the final draft.
- Shaolu Wang performed the experiments, authored or reviewed drafts of the article, and approved the final draft.
- Chunang Lian performed the experiments, authored or reviewed drafts of the article, contributed materials, and approved the final draft.
- Lisheng He conceived and designed the experiments, authored or reviewed drafts of the article, and approved the final draft.

## DNA Deposition

The following information was supplied regarding the deposition of DNA sequences:
   The snHAS sequences are available at GenBank: WWS21520.

## Data Availability

   The raw measurements are available in the Supplementary File.

## Supplemental Information

Supplemental information for this article can be found online at http://dx.doi.org/10.7717/peerj.19253#supplemental-information.

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
