# Peer review of "Expression and molecular characterization of an intriguing hyaluronan synthase (HAS) from the symbiont “Candidatus Mycoplasma liparidae” in snailfish"

_PeerJ, doi:10.7717/peerj.19253_

## Round 0.1 · original submission · Major Revisions

Authors need to address the concerns raised by the reviewers, in particular those from Reviewer 2

Reviewer 1 ·

Basic reporting

Article is well written, clear and easy to understand. article was well structured and presented well..

Experimental design

Research is in the scope of the journal. Methodology followed is well suited for the research question. Methods described with sufficient details and information.

Validity of the findings

Relevant data have been provided and it is statistically sound and controlled.Conclusion has been well supported with the presented data.

Discussion need to be enriched with some more very recent publications related with your research .

Additional comments

Some very recent references in the year 2022-24 need to be added.

Reviewer 2 ·

Basic reporting

- This manuscript describes the genetic structure and functional properties of a Hyaluronan synthase (HAS) from symbionts found within the deep-sea snailfish. Hyaluronic acids (HA) synthesized from HASs have numerous commercial applications and the potential for novel biomimetic sources are intriguing. Moreover, functional characterization of proteins, especially from species inhabiting unique ecological niches, are critical to fully appreciate biological diversity. These types of studies are too often lacking due to the labour intensive and challenging nature of expressing most proteins in vitro. However, this manuscript requires substantial editing and too many details are missing for it to be published in its current form.

- The writing needs to be improved throughout the manuscript for clearer interpretation by an international audience. For example, line 142 “At least three times of tests were performed in this study” and line 200 “further understand the evolutional relationship”

- The introduction section provides some background on Hyaluronic acid (HA) and Hyaluronan synthases (HAS) but provides almost no background on deep sea fishes, or their microbial symbionts, and why these taxa provide an ideal model system to study the molecular evolution and diversity of HASs. It is also not clear what is meant by “How to effectively and environmentally obtain HA remains a challenge” on line 73, and how this study of snail fish (or deep-sea species in general?) would help in this effort.

- The inclusion of a materials section prior at the beginning of the methods section is unconventional. Usually, specific materials are described within the methods section they are first introduced. I would advise the authors to look at other manuscripts published in PeerJ for formatting.

- The figure legends do not contain sufficient detail and in some cases the figures are not publication quality. For example, there is no description of what the colours in Figure 1 (pinks, purples) and Figure 2 (blue, red) refer to. For figure 2 there is also no mention of how the line between the points was fit, and I could not find it in the methods. In addition, it is unclear what the labels in Figure 4B refer to (presumably standards?) (ex. Data3:9600.Icd DetectorA.) I think Figure S1 is high quality and would be useful to include in the main text.

- I was able to access the HAS DNA sequence submitted to Genbank (accession number WWS21520). I also noticed the library generated for the microbial community described in the text, but it appears that this was already reported in a previous manuscript (Lian et al., 2020).

Experimental design

- While I understand that a significant amount of work is required to functionally characterization proteins, especially from deep-sea species, I cannot adequately review the validity of the results given the substantial lack of details included regarding the methods of sequence assembly and analysis.
- The unique characteristics of snHAS sequence are very interesting, but more detail is needed to understand how the authors characterized these sequences relative other HASs.
- The authors claim to have “collected and sequenced” gut microbial symbionts. What kind of sequencing was performed? Was this performed on a homogenate of the gut contents or cultured microbes? How was this “binned and assembled” for molecular cloning? Was the sequence fragment cloned amplified by Sanger sequencing or was the gene sequence uploaded to Genbank based on the data from the sequencing of the gut microbial communities in (Lian et al., 2020).
- It is unclear why the other species chosen for the sequence alignment. Were these the most similar based on a search of the entirety of the NCBI BLAST webportal? If so, what E-value cutoff was used? How does the breadth of these sequences compare to the total diversity of known HASs?
- The model used for generating the maximum likelihood tree is not reported (example HKY+I+G, JC, etc).
- It is not mentioned what software was used to evaluate the presence of transmembrane domains. This is needed to evaluate the validity of this statement on line 192.

- I do not have any background in the synthesis of HASs or the functional characterization of their activity. General readers, like myself, would require more background into what aspect of HAS function is interrogated by each test conducted
- In particular, I was unsure what the rational for introducing Cardolipin was prior to reading the discussion, making these results and Figure 3 unclear to me as I was reading the manuscript.

Validity of the findings

- The discovery and functional characterisation of a new class of HASs would be within the scope of the journal, but quite a bit more detail is required to support these findings.
- The research question as reported would benefit from substantial revision. While I believe the experiment was conducted in an appropriate manner to functionally characterize this interesting deep-sea protein, the description of the rationale used for this exploratory analysis is lacking. The paper is lacking any hypotheses testing, which may be a result of the exploratory nature of the study. However, I imagine the authors selected a deep-sea species for their study based on some underlying hypotheses on the function of proteins in the deep-sea. If more background was provided about why a deep-sea species and protein function was provided in the introduction this would become more clear.
- Methods are not described in sufficient detail to replicate the study, and as currently stated it is hard to identify how rigorous the sequence assembly and phylogenetic comparisons are.

Additional comments

I would advise the authors to review some similar recent papers in PeerJ. Some examples that might be useful for reformatting this paper are:

Barzkar et al. 2024 https://doi.org/10.7717/peerj.18149
Hillberg et al., 2023 https://doi.org/10.7717/peerj.15689

·

Basic reporting

The manuscript details the biofabrication of a novel hyaluronan synthase (HAS), along with an analysis and characterization of its kinetics. The developed enzyme shows promise for fabricating low molecular weight hyaluronic acid (HA) for various biomedical applications. However, some results require statistical analysis to strengthen the findings. Additionally, all abbreviations should be defined consistently throughout the manuscript. Below are specific suggestions to enhance the clarity and presentation of the results:

Experimental design

Abstract
• The abbreviations such as HPGLC, UDP, GlcNAc, and GlcA should be defined upon first use.
• The rationale for testing the effect of cardiolipin should be briefly mentioned.
Methodology
Sample Collection and Preparation: Additional details regarding the isolation and purification of HAS should be provides.
Statistical Analysis: Introduce a section outlining the statistical methods used to analyze key results. Specifically, include analysis of changes in HA concentration over time in response to enzymatic activity, and the effect of cardiolipin on HAS activity.
Results
Figure 4: Colour indicators for each peak in Figure 4B should be added to either the legend or directly on the figure for easier interpretation. The retention times for all peaks should be included either within the figure or in a supplementary table.

Validity of the findings

All issues were included in the previous section

---

## Round 0.2 · Minor Revisions

Please address comments raised by Reviewer 2.

**Language Note:** The review process has identified that the English language must be improved. PeerJ can provide language editing services - please contact us at [email protected] for pricing (be sure to provide your manuscript number and title). Alternatively, you should make your own arrangements to improve the language quality and provide details in your response letter. – PeerJ Staff

Reviewer 1 ·

Basic reporting

Authors edited the manuscript appropriately as per the suggestions from the review. Added latest relevant references in the text. Writing has been improved with clear professional English. Now manuscript is ready to be published.

Experimental design

Methodology has been further improved and satisfied the reviews query. It is now clear and satisfactory for the readers.

Validity of the findings

Result section has also been improved. Figure legends are now much more clear with more details about the figures. Data has been well presented and justified with supportive references.

Additional comments

Nil

Reviewer 2 ·

Basic reporting

The writing throughout the manuscript has been improved. However, there are still quite a few sentences throughout that could use some re-writing to avoid any ambiguity. Some examples are included below.
Lines 108-109 Why does it have a “recombinant plasmid”? And why is this important? These methods need to be stated more clearly.
Line 161: “triploid” refers to genetic disorders. You could use “three volumes” or “three times the volume”, or “triple the volume”.
Lines 179-180: “in snHAS, only the first 88D and 208Q residues were conserved (Fig. 1A).” I would reword to “in snHAS, only the first two residues, 88D and 208Q, were conserved (Fig. 1A).” or it sounds like hundreds of residues are not conserved.
Lines 307 “than previously reported ones” is not specific enough. What are you referring to as “ones”.
--
The figures have also been improved. Some further minor comments below.
Figure 3: It needs to be stated what method was used to draw the lines between the points. Is this Loess smoothing or some other method? The line could also be removed. I have not heard colour referred to as “point” before. Maybe rephrase to “the reaction with hyaluronidase added at 60 mins is depicted in red.”
I do not know how to interpret “The progress was performed as described in methods”; this statement needs to be re-written for clarity.
Figure 5. I would recommend aligning the x-axes in the two plots so that it is easier to compare the HAs to the standards.
--
Some additional context is needed in the introduction, methods/results and discussion.
Line 84: “Obtaining HA in an effective and environmentally friendly manner remains a challenge” This statement needs some more support, it currently does not have any references associated with it. What are some of the environmentally damaging processes currently used? Similarly, not much detail is provided in the discussion on lines 293-294 where this topic is also mentioned. Also note that the name of the article cited here, Jin et al., 2016, is incorrect in the references.
Lines 214-216. It is not totally clear why the E173D mutant was tested. The authors should include some description why this mutant was made and why it is believed to be important to the HA synthesis somewhere in the methods or results.
Since this is a symbiont. I would add a statement in the deep-sea paragraph about how it is not only deep-sea species that need to adapt, but also species that associated with deep-sea species. This would set up the study system better.

Experimental design

The work is within the scope of PeerJ.
The research question has been greatly improved from the previous version of the manuscript but could still use a bit of refinement. In particular, I think the authors should directly state why they are interested in studying a deep-sea species HAS activity. It is largely exploratory in nature as currently written.
Methods and results are now more clearly presented.
The authors additions to the methods have addressed my major concerns regarding the rigor and repeatability of this manuscript. Some minor points are included in previous section.

Validity of the findings

The experimental data presented in this study are robust. Few statistical analyses are presented.
Conclusions are appropriate.

---

## Round 0.3 · accepted · Accept

Authors have addressed all reviewers' comments